# Upper-Arm SBP Decline Associated with Repeated Cuff-Oscillometric Inflation Significantly Correlated with the Arterial Stiffness Index

**DOI:** 10.3390/jcm11216455

**Published:** 2022-10-31

**Authors:** Noriyuki Kawaura, Rie Nakashima-Sasaki, Hiroshi Doi, Kotaro Uchida, Takuya Sugawara, Sae Saigo, Kaito Abe, Kentaro Arakawa, Koichi Tamura, Kiyoshi Hibi, Tomoaki Ishigami

**Affiliations:** 1Department of Medical Science and Cardiorenal Medicine, Yokohama City University Graduate School of Medicine, Yokohama 236-0004, Japan; 2Kawasaki Saiwai Clinic, Kawasaki 212-0016, Japan; 3Department of Cardiology, Kanagawa Cardiovascular and Respiratory Center, Yokohama 236-0051, Japan; 4Department of Cardiology, Yokohama City Medical Center, Yokohama 232-0024, Japan

**Keywords:** arterial stiffness, atherosclerosis, non-invasive, cuff-oscillometric, repeated measurements, blood pressure change

## Abstract

We evaluated the clinical significance of the new non-invasive vascular indices to explore their potential utility using repeated cuff-oscillometric inflation. In 250 consecutive outpatients, we performed a cross-sectional, retrospective, single-center, observational study to investigate sequential differences in arterial stiffness using blood pressure, arterial velocity pulse index (AVI), and arterial pressure volume index (API) with repeated measurements. Males accounted for 62.7% of the patients, and the mean age was 68.1 ± 12.1 years. The mean systolic blood pressure (SBP) and diastolic blood pressure (DBP) of the first reading in repeated measurements were 133.07 ± 21.20 mmHg and 73.94 ± 13.56 mmHg, respectively. The mean AVI and API were 23.83 ± 8.30 and 31.12 ± 7.86, respectively. In each measurement of these parameters, although DBP and AVI did not show significant changes throughout repeated measurements, SBP and API decreased significantly according to the measurement orders. Furthermore, changes in SBP and API were significantly correlated in several of the models. In this study, it was concluded that upper-arm SBP decline associated with repeated cuff-oscillometric inflation was significantly correlated with the arterial stiffness index. The findings of this study will allow clinicians to easily recognize the progression of atherosclerosis through regular, routine practice. In conclusion, this study suggests that changes in repeated SBP measurements may be predictive of arterial stiffness and atherosclerosis.

## 1. Introduction

Atherosclerosis is the common result of the advanced progression of diseases based on diet and exercise in daily life, such as hypertension (HT), diabetes, dyslipidemia (DL), or lifestyle-related diseases. Consequently, atherosclerosis can lead to fatal cardiovascular events, and it is necessary to salvage and counteract life-threatening diseases such as acute myocardial infarction and acute cerebral infarction. Even if fatality can be avoided, it is a strong limiting factor for healthy life expectancy as it can cause serious sequelae with disability and result in the need for permanent nursing care [1,2,3].

To date, the mainstream countermeasures against atherosclerotic events have been the establishment of a life-saving emergency system that concentrates medical resources on the disease and its symptoms that develop in an acute catastrophe and the control of population risk through public and personal hygiene by controlling blood pressure (BP), blood sugar, and lipids, which are risk factors for the disease. The former requires not only enormous social costs, including human and medical resources and costs, for unpredictable lethal events but also poses other challenges, such as regional disparities among nationwide healthcare services that can and cannot supply them and inappropriate work overload for healthcare personnel [4,5]. The latter is expected to have a uniform effect with no regional differences; however, its results for medical care may take a long time to achieve, and there is also the problem of lack of certainty. Therefore, since atherosclerosis is a disease caused by an abnormal biological process, its control and suppression require both molecular and cellular level understanding of its pathogenesis and technological innovations [6]. Current oncology has made substantial progress with a successful inventory of various molecular target biologics [7]. Additionally, earlier diagnosis and detection of unaware subclinical atherosclerosis with less invasive clinical devices are necessary to adequately monitor disease progression, which is necessary for novel drug discovery in these areas [3].

Recent technological innovations have made it possible for us to use the AVE-1500 (Shisei Datum, Tokyo, Japan), which can easily measure indices reflecting arterial stiffness and central arterial pressure with little physical burden using the cuff-oscillometric method. The AVE-1500 can measure the arterial pressure volume index (API), an index reflecting arterial stiffness, and the arterial velocity pulse index (AVI), an index reflecting central arterial pressure. Several studies have clarified the relationship between these indices and arterial stiffness [8,9].

HT is one of the most important lifestyle-related diseases that causes atherosclerosis. Guidelines for the management of HT have been established by various academic societies worldwide. It is frequently observed that repeated measurement of BP changes its value; therefore, the guidelines recommend using an average of two or three measurements on a single occasion [10,11,12]. Repeated cuff-oscillometric inflation is necessary to measure the BP two or three times on a single occasion, and it is frequently noted that BP changes were observed throughout the measurements. Our analyses aimed to reveal the regularity of BP changes with repeated cuff-oscillometric inflation at BP measurements.

It is unknown how repeated cuff-oscillometric inflations with BP measurement are related to arterial stiffness and central arterial pressure and how changes to the API and AVI will be brought about by repeated cuff-oscillometric inflations. Therefore, in the present study, we evaluated the clinical significance of the new non-invasive vascular indices, AVI and API, in outpatients with various clinical backgrounds to explore the potential utility of these two indices by using repeated cuff-oscillometric inflation in actual clinical settings.

## 2. Materials and Methods

### 2.1. Study Design and Population

We performed a cross-sectional, retrospective, observational study at the Yokohama City University Hospital in Japan. The study protocol was registered and approved by the ethics committee of Yokohama City University Hospital in 2015 (B150701005) with notifications for guaranteed withdrawal of participants on the website providing means of ‘‘opt-out,” and due to the non-invasive observational study design, we did not request additional informed consent from the participants.

We used a multifunctional BP monitoring device, AVE-1500 (Shisei Datum, Tokyo, Japan), to evaluate the AVI and API in 250 consecutive outpatients in the Department of Medical Science and Cardiorenal Medicine between May 2013 and March 2015. Patients who had atrial fibrillation were excluded from the study. Medical records were reviewed to collect data for each patient’s profile, general status, medical history, laboratory data, and concomitant medications. HT was defined as systolic BP (SBP) of ≥140 mmHg, diastolic BP (DBP) of ≥90 mmHg, or ongoing medical treatment for HT. DL was defined as a low-density lipoprotein (LDL) cholesterol level of ≥140 mg/dl, triglyceride level of ≥150 mg/dl, high-density lipoprotein (HDL) cholesterol level of ≤40 mg/dl, or ongoing medical treatment for DL. Chronic heart failure was defined as a B-type natriuretic peptide level of ≥40 pg/mL caused by cardiovascular disease. Valvular heart disease was defined as valve regurgitation/stenosis of at least a moderate degree. Plasma glucose and triglyceride levels were measured using casual blood sampling without overnight fasting.

### 2.2. AVI and API

Both AVI and API are novel arterial indices measurable by a multifunctional BP monitoring device, AVE-1500 (Shisei Datum, Tokyo, Japan), with cuff-oscillometric-based technologies. The AVI is an index of the characteristics of the pulse waveform at high cuff pressures above the maximum BP. Recent findings have revealed that central arterial stiffness and characteristics are strongly reflected in the pulse wave in this pressure region [8,9]. This pulse wave has an increased late systolic waveform and a steeper falling curve thereafter, owing to an increase in the reflected wave due to aging, stiffening, and increased peripheral resistance. However, the ejection phase is not affected by the reflected wave, and as a result, the pulse wave amplitude differentiated from the ejection phase increases only the velocity change during brachial artery diastole (cardiac systole) out of the absolute value of the bottom of the valley of differentiated waveforms between pulse wave and time (Vr) and velocity change during brachial artery relaxation (cardiac diastole), and the value of the pulse waveform characteristics (Vr/first peak of the differentiated waveform between pulse wave and time (Vf)) become an index that varies with the magnitude of the reflected wave [13].

The API is an index of arterial pressure–volume characteristics derived from the arterial volume change resulting from cuff pressure relative to the internal and external pressure difference (BP–cuff pressure) applied to the arterial wall. Previous studies have shown that softer vessels show more rapid changes in arterial volume with changes in cuff pressure and that the degree of the slope of this curve varies primarily with the stiffness of the vessel’s media and adventitia. API has developed and indexed a method to stably evaluate the differences in this characteristic [9].

The AVE-1500 (Shisei Datum, Tokyo, Japan) is a newly developed device that can non-invasively evaluate arterial stiffness and endothelial dysfunction of the central arteries (AVI) and peripheral arteries (API) using a cuff-oscillometric method in a single BP measurement. AVI and API were measured using an AVE-1500 with the participants in the supine position. Finally, the AVE-1500 can evaluate the conventional SBP, DBP, API, AVI, and pulse rate in a single measurement.

Measurements were taken at least thrice for each participant in a quiet temperature-controlled room (24.0–26.8 °C). The average measurements for API and AVI at the time of participant enrollment were used for subsequent analyses.

### 2.3. Statistical Analysis

Data are shown as the mean ± standard deviation for continuous variables and as frequencies and percentages for categorical variables. Analysis for time-series measurements was compared using repeated measures analysis of variance and Friedman rank sum test, with additional analysis using the Holm method. Statistical significance was set at *p* < 0.05. The relationships between AVI and API and all other variables were analyzed using a linear regression model and Pearson’s correlation coefficient. Statistical analysis was performed using R version 4.1.2 (The R Foundation for Statistical Computing, Vienna, Austria).

## 3. Results

### 3.1. Baseline Characteristics

Of the 250 enrolled participants, 236 who met the inclusion criteria and did not have missing key data were analyzed. The baseline patient characteristics are shown in Table 1. Of the patients, 62.7% were males, with an overall mean age of 68.1 ± 12.1 years. This study included 158 patients with HT (67.0%), 48 with diabetes (20.3%), and 109 with DL (46.2%). The mean creatinine (Cr) and estimated glomerular filtration rate were 0.98 ± 1.01 mg/dl and 69.26 ± 23.72 mL/min/1.73 m^2^, respectively. In Table 2, the antihypertensive drugs were renin–angiotensin system inhibitors (121 patients, 51.3%), calcium-channel antagonists (119 patients, 50.4%), β-blockers (71 patients, 30.1%), and diuretics (54 patients, 22.9%). The mean SBP and DBP of the first measurement were 133.1 ± 21.2 mmHg and 73.9 ± 13.6 mmHg, respectively (Table 3). The mean AVI and API of the first measurement were 23.8 ± 8.3 and 31.1 ± 7.9, respectively. The total number of BP, AVI, and API measurements was 2452; therefore, an average of 9.89 measurements per person were performed.

### 3.2. Changes in BP, API, and AVI with Repeated Measurements

The results of the repeated measurements of BP (SBP/DBP), AVI, and API are shown in Table 4 and Figure 1. The average of the first SBP measurement in repeated measurements (SBP01) was 136.6 mmHg, the second (SBP02) was 133.6 mmHg, and the third (SBP03) was 131.9 mmHg. The average of each measurement of DBP, AVI, and API is also shown in Table 4. In each measurement, although DBP and AVI did not show significant changes throughout the measurements, SBP and API decreased significantly according to the measurement orders, as shown in Figure 1.

### 3.3. Regression Analysis of the Relationship between Novel Vascular Indices and Clinical Characteristics

Univariate and multivariate analyses of SBP change, defined as “∆SBP = (SBP01 − SBP03)/SBP01, were conducted to determine the association between each clinical parameter and measure (Table 5 and Table 6). In the univariate analysis of the SBP change, sex and Cr were significant in this model, and in the multivariate analysis, sex was significant (Table 5). The same model was adapted for the API. Regarding the API change, in the univariate analysis, age, diabetes mellitus (DM), sex, and Cr were significant in this model, whereas in the multivariate analysis, only age, DM and Cr were significant (Table 6). ∆SBP and ∆API were significantly correlated (Figure 2). Moreover, another three models (models 1 and 2: ∆SBP = (SBP01 − SBP03)/SBP01, models 3 and 4: ∆SBP = (SBP01 − SBP03)/mean SBP, models 1 and 3 included hemodialysis patients; models 2 and 4 did not include hemodialysis patients) were created and analyzed to ensure that the analysis was accurate for the phenomenon that occurred. In the univariate analysis of the changes in SBP, sex, Cr, smoker and LDL were significant in some models, and in the multivariate analysis, LDL was significant in one model and sex was significant in all models. Regarding the changes in API, in the univariate analysis, age, diabetes mellitus (DM), sex, Cr, and ischemic heart disease (IHD) were significant in some models, whereas in the multivariate analysis, only age and DM were significant in all models. In all models, ∆SBP and ∆API were significantly correlated.

## 4. Discussion

The AVE-1500 is a newly developed medical device that can simultaneously measure BP and evaluate peripheral arterial stiffness and central arterial pressure [14,15]. In the present study, we used the AVE-1500 to measure API, an index of peripheral arterial stiffness, and AVI, an index of central arterial pressure, not in a single measurement but in repeated BP measurements, and showed the relationship between these changes. We also investigated the relationship between these changes and each of the parameters associated with atherosclerosis. In the repeated measurements of BP and API/AVI, the changes in SBP and API showed a similar regression that decreased by measurement thrice. In the univariate analysis of the changes in SBP, sex, smoking, Cr, and LDL were significant in some models, and in the multivariate analysis, LDL was significant in one model and sex was significant in all models. Regarding the changes in API, in the univariate analysis, age, DM, sex, Cr, and IHD were significant in some models, while in the multivariate analysis, only age and DM were significant in all models.

Several studies have reported changes with repeated BP measurements on a single occasion. Some reports have suggested that the second measurement is lower than the first, and the third measurement is lower than the second [16,17]. Other reports have indicated that the measurement interval has some influence on changes in repeated BP measurements [18]. Our study also showed a significant stepwise decrease in the first, second, and third measurements, similar to previous reports. On the other hand, some reports have suggested that BP variability can be attributed to psychological effects and regression to the mean, in addition to baroreflex effects [19,20]. The current guidelines for HT in most societies recommend measuring BP at least twice on a single occasion [10,11,12].

In previous studies that investigated the relationship between BP and API/AVI on single measurements, almost all the studies we examined showed significant correlations between SBP and API [21,22,23,24]. In most of these studies, the strength of the correlation was moderate, while some studies showed a strong correlation. Although regression analysis has also shown significant results in several studies, all of them were based on single measurements. The correlation between SBP and API in a single measurement was also significant in the present study. Moreover, to further explore the relationship between these two variables, we focused on the changes in SBP and API, which showed a linear decreasing type with statistical significance in repeated measurements, and analyzed the relationship between them. To analyze the changes in SBP and API, two models were used: one in which the change was divided by the first measurement and the other in which the change was divided by the average of the three measurements to ensure that the analysis was accurate for the phenomenon that occurred. The results showed that in both models, there was a significant correlation between the changes in SBP and API.

The results of the analyses of the relationship between the changes in API and SBP with repeat measurements and some variables in the four models showed that SBP and API were related to variables known to be associated with arterial stiffness and atherosclerosis. Although previous studies have examined SBP and API using single measurements (not repeated measurements), significant correlations were noted for age, the prevalence of diabetes, smoking, and sex, and the results were consistent with those of the present study [9,21,22,23].

Although there have been several previous reports on the comparison of BP and API/AVI with risk factors for atherosclerosis in single measurements, to the best of our knowledge, this is the first report of a study investigating the relationship between BP and API/AVI with changes in repeated measurements. Prior to the development of the AVE-1500, BP and vascular stiffness indices such as API/AVI were evaluated separately with different devices to measure them; therefore, the relationship between them was undetermined. However, with the new development of the AVE-1500 BP and pulse wave meter, both can now be compared simultaneously, making it possible to better determine the relationship between BP and central artery index and arterial stiffness. Although the relationship between SBP and API in a single measurement has already been reported in previous studies because of the availability of simultaneous measurements, we confirmed that the relationship was stronger by adding evaluation with repeated measurements in the present study. The reason for the strong relationship between SBP and API may be that API is an index of arterial stiffness, as mentioned above, as well as an index of arterial volume change from the viewpoint of the measurement principle; therefore, it is appropriate from the measurement principle that the changes in API show a strong relation to the changes in SBP. In other words, this study suggests that changes in repeated SBP measurements may be predictive of arterial stiffness and atherosclerosis. However, we cannot conclude whether “SBP decline with repeated cuff-oscillometric pressure” can be used as a substitute for arterial stiffness in this study alone, and we can only propose a hypothesis. It is necessary to examine it from a different perspective, for example, by comparing it with other stiffness modalities.

The present study has some limitations. The study was cross-sectional and retrospective, and the sample size was relatively small. In addition, the participants of this study were outpatients of a cardiorenal division. Therefore, future research is required, and we believe that this study only proves the concept of a voluminous subject. However, we have concluded that we might be able to take arterial stiffness and stiffness one step further in the present study by using the new device AVE-1500. The authors should discuss the results and how they can be interpreted from the perspective of previous studies and the working hypotheses. These findings and their implications should be discussed in the broadest possible context. Future research directions may also be highlighted.

## 5. Conclusions

In conclusion, this study suggests that changes in repeated SBP measurements may be predictive of arterial stiffness and atherosclerosis.

## Figures and Tables

**Figure 1 jcm-11-06455-f001:**
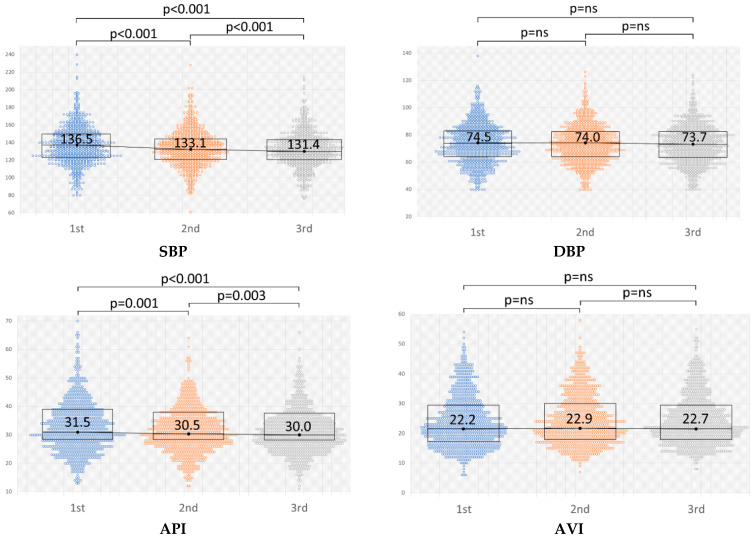
Changes in the repeated measurements of blood pressure (BP) (SBP/DBP), AVI, and API. The value of BP is represented in mmHg. Significant changes are indicated by *p* < 0.05. SBP, systolic blood pressure; DBP, diastolic blood pressure; AVI, arterial velocity pulse index; API, arterial pressure volume index.

**Figure 2 jcm-11-06455-f002:**
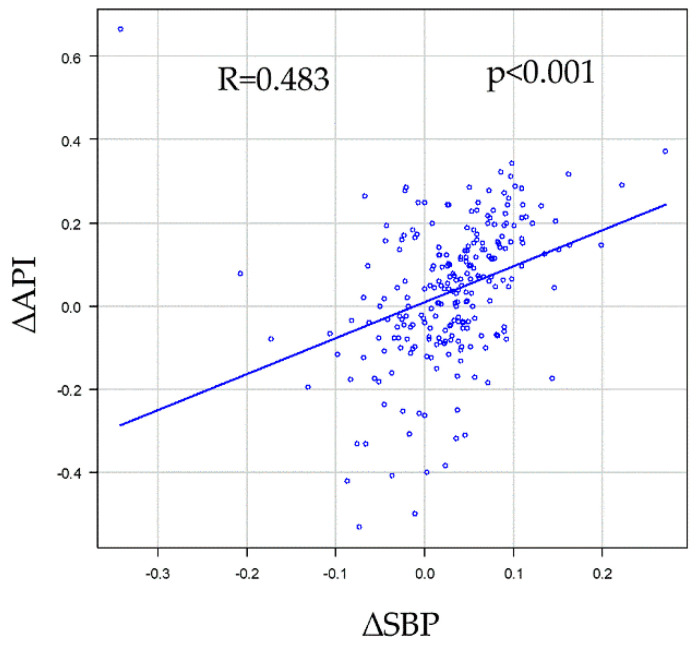
Correlations between SBP change (∆SBP) and API change (∆API). ∆SBP was defined as (SBP01 − SBP03)/SBP01, and the same model was adapted for the API (∆API = (API01 − API03)/API01). ∆SBP and ∆API significantly correlated (R = 0.483, *p* < 0.001). API, arterial pressure volume index; SBP, systolic blood pressure; SBP01 (API01), first SBP (API) measurement in repeated measurements; SBP03 (API03), third SBP (API03) measurement in the repeated measurements.

**Table 1 jcm-11-06455-t001:** Baseline characteristics.

	Male (*n* = 148)	Female (*n* = 88)	Total (*n* = 236)
Age (years)	68.0 ± 12.1	68.2 ± 12.1	68.1 ± 12.1
Hypertension (%)	99 (66.9)	59 (67.1)	158 (67.0)
Diabetes (%)	38 (25.7)	10 (11.4)	48 (20.3)
Dyslipidemia (%)	70 (47.3)	39 (44.3)	109 (46.2)
IHD (%)	28 (18.9)	6 (6.8)	34 (14.4)
ASO (%)	11 (7.4)	4 (4.6)	15 (6.4)
VHD (%)	11 (7.4)	10 (11.4)	21 (8.9)
OMI (%)	12 (8.1)	4 (4.6)	16 (6.8)
CHF (%)	20 (13.5)	10 (11.4)	30 (12.7)
Cardiomyopathy (%)	5 (3.4)	4 (4.6)	9 (3.8)
PH (%)	1 (0.7)	3 (3.4)	4 (1.7)
AF (%)	29 (19.6)	7 (8.0)	36 (15.3)
PM/ICD (%)	2 (1.4)	0 (0)	2 (0.85)
COPD (%)	5 (3.4)	0 (0)	5 (2.1)
Smoking			
Current (%)	24 (16.2)	7 (8.0)	31 (13.2)
Past (%)	61 (41.2)	6 (6.8)	67 (28.4)
Never (%)	63 (42.6)	75 (85.2)	138 (58.4)
Laboratory data			
Creatinine (mg/dL)	1.12 ± 1.20	0.73 ±0.46	0.98 ± 1.01
eGFR (mL/min/1.73 m^2^)	67.45 ± 21.96	72.43 ± 26.34	69.26 ± 23.72
Uric acid (mg/dL)	5.90 ± 1.46	5.01 ± 1.35	5.58 ± 1.49
Plasma glucose (mg/dL)	130.29 ± 34.35	116.92 ± 32.07	125.84 ± 34.07
HbA1c (%)	6.10 ± 0.84	6.15 ± 1.00	6.11 ± 0.89
CRP (mg/dL)	0.32 ± 0.75	0.49 ± 1.25	0.38 ± 0.96
TG (mg/dL)	143.63 ± 80.09	152.28 ± 95.43	146.49 ± 85.35
HDL-C (mg/dL)	59.10 ± 16.30	66.58 ± 22.38	61.54 ± 18.79
LDL-C (mg/dL)	105.27 ± 34.80	115.47 ± 35.78	108.85 ± 35.38
T-Cho (mg/dL)	181.37 ± 37.64	204.55 ± 44.24	189.29 ± 41.36
BNP (pg/mL)	121.06 ± 230.01	62.6 ± 106.4	99.93 ± 196.27

Data are presented as the mean ± standard deviation or *n* (%). IHD, ischemic heart disease; ASO, arteriosclerosis obliterans; VHD, valvular heart disease; OMI, old myocardial infarction; CHF, chronic heart disease; PH, pulmonary hypertension; AF, atrial fibrillation; PM/ICD, pacemaker/implantable cardioverter-defibrillator; COPD, chronic obstructive pulmonary disease; eGFR, estimated glomerular filtration rate; HbA1c, glycosylated hemoglobin; CRP, C-reactive protein; TG, triglyceride; HDL-C, high-density lipoprotein cholesterol; LDL-C, low-density lipoprotein cholesterol; T-Cho, total cholesterol; BNP, brain natriuretic peptide.

**Table 2 jcm-11-06455-t002:** Medications at baseline.

	Male (*n* = 148)	Female (*n* = 88)	Total (*n* = 236)
RAS inhibitors (%)	79 (53.4)	42 (47.7)	121 (51.3)
Ca antagonists (%)	72 (48.7)	47 (53.4)	119 (50.4)
β-blockers (%)	48 (32.4)	23 (26.1)	71 (30.1)
Diuretics (%)	32 (21.6)	22 (25.0)	54 (22.9)
α-blockers (%)	1 (0.7)	1 (1.1)	2 (0.9)
Nitrites (%)	16 (10.8)	8 (9.1)	24 (10.2)
Biguanides (%)	7 (4.7)	3 (3.4)	10 (4.2)
Statins (%)	60 (40.5)	39 (44.3)	99 (42.0)
Bezafibrates (%)	7 (4.7)	2 (2.3)	9 (3.8)
EPA (%)	6 (4.1)	1 (1.1)	7 (3.0)
Ezetimibe (%)	6 (4.1)	3 (3.4)	9 (3.8)
Sulfonylureas (%)	11 (7.4)	1 (1.1)	12 (5.1)
α-glucosidase inhibitors (%)	13 (8.8)	0 (0)	13 (4.3)
Thiazolidine (%)	6 (4.1)	1 (1.1)	7 (3.0)
DPP-4 inhibitors (%)	17 (11.5)	6 (6.8)	23 (9.8)
GLP-1 analogs (%)	2 (1.4)	0 (0)	2 (0.9)
Insulin (%)	8 (5.4)	2 (2.3)	10 (4.2)
Aspirin (%)	46 (31.1)	18 (20.5)	64 (27.1)
Thienopyridine (%)	15 (10.1)	3 (3.4)	18 (7.6)
Cilostazol (%)	6 (4.1)	4 (4.6)	10 (4.2)
Sarpogrelate (%)	1 (0.7)	0 (0)	1 (0.4)
Dipyridamole (%)	0 (0)	0 (0)	0 (0)
Prostaglandin (%)	3 (2.0)	1 (1.1)	4 (1.7)

Data are presented as the mean ± standard deviation or *n* (%). RAS, renin-angiotensin system; Ca, calcium; EPA, eicosapentaenoic acid; DPP-4, dipeptidyl-peptidase-4; GLP-1, glucagon-like peptide-1.

**Table 3 jcm-11-06455-t003:** Measurements of Blood pressure, AVI and API at baseline.

	Male (*n* = 148)	Female (*n* = 88)	Total (*n* = 236)
SBP (mmHg)	131.9 ± 19.8	135.2 ± 23.4	133.1 ± 21.2
DBP (mmHg)	75.3 ± 34.4	71.7 ± 14.4	73.9 ± 13.6
AVI	22.9 ± 8.4	25.4 ± 7.9	23.8 ± 8.3
API	29.3 ± 6.9	34.2 ± 8.5	31.1 ± 7.9
Pulse rate (/min)	73.5 ± 12.4	75.2 ± 14.4	74.1 ± 13.2
Measurements of AVI and API			
Total number	1545	907	2452
Total visits	300	496	796
Per visit per person	3.02	3.12	3.04

Data are presented as the mean ± standard deviation or *n* (%). SBP, systolic blood pressure; DBP, diastolic blood pressure; AVI, arterial velocity pulse index; API, arterial pressure volume index.

**Table 4 jcm-11-06455-t004:** The average of each measurement in repeated measurements.

	First Measurement	Second Measurement	Third Measurement
SBP	136.6± 20.4	133.6 ± 19.8	131.9 ± 18.2
DBP	74.5 ± 13.8	74.0 ± 12.7	73.7 ± 13.0
AVI	23.8 ± 8.2	24.1 ± 7.8	24.2 ± 7.7
API	32.5 ± 8.8	31.1 ± 6.5	30.5 ± 6.4

SBP, systolic blood pressure; DBP, diastolic blood pressure; AVI, arterial velocity pulse index; API, arterial pressure volume index.

**Table 5 jcm-11-06455-t005:** Multivariate regression results with SBP change.

Independent Variables	Standardized Regression Estimate	95%CI		*p*
Sex	0.0261	0.0081	0.0440	0.005 *
Creatinine	−0.0058	−0.0134	0.0019	0.14

Multivariate regression results with SBP change were analyzed. SBP change was defined as (SBP01 − SBP03)/SBP01. The adopted independent variables were significant in the univariate regression analysis. Hypertension was excluded from the analysis because of its presumed strong relationship with SBP. Adjusted R-squared 0.044, F-statistic 6.238, *p*-value 0.002. SBP, systolic blood pressure; SBP01 (03), first (third) SBP measurement in the repeated measurements. * *p* < 0.05.

**Table 6 jcm-11-06455-t006:** Multivariate regression results with API change.

Independent Variables	Standardized Regression Estimate	95%CI		*p*
Age	−0.0020	−0.0038	−0.0003	0.020 *
Diabetes	−0.0578	−0.1105	−0.0050	0.032 *
Sex	0.0248	−0.0200	0.0695	0.276
Creatinine	−0.0189	−0.0378	−0.00001	0.050

Multivariate regression results with API change were analyzed. API change was defined as (API01 − API03)/API01. The adopted independent variables were significant in the univariate regression analysis. Adjusted R-squared 0.060, F-statistic 4.63, *p*-value 0.001. API, arterial pressure volume index; API01 (03), first (third) API measurement in the repeated measurements. * *p* < 0.05.

## Data Availability

The data presented in this study are available upon request from the corresponding author. The data are not publicly available due to privacy regulations.

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
