# Peer review of "Upper-Arm SBP Decline Associated with Repeated Cuff-Oscillometric Inflation Significantly Correlated with the Arterial Stiffness Index"

_jcm, 2022, doi:10.3390/jcm11216455_

Round 1

Reviewer 1 Report

This study showed the effect of changes in the repeated measurement of BP on the results of API and AVI. A total of 2452 measurements of BP, AVI, and API were performed on 236 patients, and the authors say that 9.89 measurements were performed per person. In addition, blood pressure was measured three times in single occasion, and API and AVI results were measured through AVE-1500. Here, the authors insisted that there is a correlation between SBP and API. In this study, since API and AVI measured by AVE-1500 are correlated with pressure waves, there is sufficient evidence that API and SBP show a linear correlation. However, the present study concluded "changes in repeated SBP measurements may be predictive of arterial stiffness and atherosclerosis." There is no evidence for the effect of correlation on the prediction of arterial stiffness and atherosclerosis regarding the variation of API, and the explanation is insufficient. The point is that the correlation between SBP and API suggests that API value is more affected by SBP, but there is a problem in claiming that arterial stiffness and atherosclerosis are correlated with repeated SBP measurement with the design of this study.

There are three Table 1 as a minor comment. It would be better to divide and present. It is recommended to provide a flow chart for retrospective analysis to enhance understanding.

Author Response

Attached please find our response to the reviewer #1 in pdf form.

Reviewer 2 Report

The authors compared changes of blood pressure (BP) with multiple BP measurements with changes in arterial stiffness indices in 250 consecutive outpatients. The conclude that changes in systolic BP are associated with arterial stiffness indices. This is per se a very important topic. However, the methodology with multiple models is difficult to understand. I only found models with the CHANGE of arterial pressure volume index (API) to be associated with multiple BP measurements. To clear the difficult methodology, I would recommend to use just one multivariate model and with ABSOLUTE API as dependent variable. This way, (if the result is positive), the authors could conclude that systolic BP decline by repeated BP measurements could be used for atherosclerosis detection without the need for a special device that can measure vascular indices non-invasively. The manuscript is written in good English. Citations seem ok.

Minor comments:

-        In my opinion it would be favourable to remove the second decimal from the blood pressure values throughout the manuscript.

-        Table 3-2: 95% CIs should be presented.

Author Response

Attached please find our response to the reviewer #2 in pdf.

Round 2

Reviewer 2 Report

I have no further comments.